# Rumen Development of Tianhua Mutton Sheep Was Better than That of Gansu Alpine Fine Wool Sheep under Grazing Conditions

**DOI:** 10.3390/ani14091259

**Published:** 2024-04-23

**Authors:** Dengpan Li, Zhanjing Liu, Xinming Duan, Chunhui Wang, Zengping Chen, Muyang Zhang, Xujie Li, Youji Ma

**Affiliations:** 1College of Animal Science and Technology, Gansu Agricultural University, Lanzhou 730070, China; ldp7208@163.com (D.L.); liuzhanjin@126.com (Z.L.); chwang5522@163.com (C.W.); 17352055872@163.com (Z.C.); zmy9905222023@163.com (M.Z.); lxj804599@126.com (X.L.); 2Gansu Key Laboratory of Animal Generational Physiology and Reproductive Regulation, Lanzhou 730070, China; 3Tianzhu County Animal Disease Prevention and Control Center, Wuwei 733200, China; 4NongfaYuan Zhejiang Agricultural Development Co., Ltd., Huzhou 313000, China; llll5252@126.com

**Keywords:** Tianhua mutton sheep, Gansu alpine fine wool sheep, rumen microbiota, Hematoxylin Eosin, volatile fatty acids

## Abstract

**Simple Summary:**

The development of the sheep industry plays an important role in increasing the income of farmers and herdsmen and promoting rural revitalization. Genetic improvement is the key to improving the competitiveness of the sheep industry. Tianhua mutton sheep, as China’s first mutton fine wool sheep adapted to alpine pastures, have a key advantage over Gansu alpine fine wool sheep in terms of adaptive ability. In order to show the differences in rumen development and microbial structure between Tianhua mutton sheep and Gansu alpine fine wool sheep, the present study was conducted to comprehensively investigate the rumen of the two types of fine wool sheep by Hematoxylin Eosin (H&E) staining, volatile fatty acids (VFAs) assay, and full-length sequencing of 16S rRNA. The results showed that Tianhua mutton sheep had a better performance in the rumen volatile fatty acid content, rumen tissue development, and rumen microorganisms compared with Gansu alpine fine wool sheep, and this study provided a good theoretical explanation for the better adaptation of Tianhua mutton sheep to alpine pastures compared with Gansu alpine fine wool sheep.

**Abstract:**

The purpose of this experiment was to investigate the differences in rumen tissue morphology, volatile fatty acid content, and rumen microflora between Tianhua mutton sheep and Gansu alpine fine wool sheep under the same grazing conditions. Twelve 30-day-old lambs were randomly selected from two different flocks in Duolong Village and grazed together for a period of 150 days. The rumen tissue was fixed with 4% paraformaldehyde and brought back to the laboratory for H&E staining, the volatile fatty acid content of the rumen contents was detected by gas chromatography, and the rumen flora structure was sequenced by full-length sequencing of the bacterial 16S rRNA gene using the PacBio sequencing platform. The acetic acid and total acid contents of the rumen contents of Tianhua mutton sheep were significantly higher than those of Gansu alpine fine wool sheep (*p* < 0.05). The rumen papillae height of Tianhua mutton sheep was significantly higher than that of Gansu alpine fine wool sheep (*p* < 0.05). The diversity and richness of the rumen flora of Tianhua mutton sheep were higher than those of Gansu alpine fine wool sheep, and Beta analysis showed that the microflora structure of the two fine wool sheep was significantly different. At the phylum level, Firmicutes and Bacteroidetes dominated the rumen flora of Tianhua mutton sheep and Gansu alpine fine wool sheep. At the genus level, the dominant strains were *Christensenellaceae_R_7_group* and *Rikenellaceae_RC9_gut_group*. LEfSe analysis showed that *Prevotella* was a highly abundant differential species in Tianhua mutton sheep and lachnospiraccac was a highly abundant differential species in Gansu alpine fine wool sheep. Finally, both the KEGG and COG databases showed that the enrichment of biometabolic pathways, such as replication and repair and translation, were significantly higher in Tianhua mutton sheep than in Gansu alpine fine wool sheep (*p* < 0.05). In general, there were some similarities between Tianhua mutton sheep and Gansu alpine fine wool sheep in the rumen tissue morphology, rumen fermentation ability, and rumen flora structure. However, Tianhua mutton sheep had a better performance in the rumen acetic acid content, rumen papillae height, and beneficial bacteria content. These differences may be one of the reasons why Tianhua mutton sheep are more suitable for growing in alpine pastoral areas than Gansu alpine fine wool sheep.

## 1. Introduction

The number of sheep in Gansu Province rank third in the country, which is inseparable from the role of excellent local breeds. As the traditional grassland animal husbandry area, the healthy and sustainable development of animal husbandry in the alpine pastoral area can be promoted by further optimizing the herd structure through the improvement project of mutton sheep breeds. Gansu alpine fine wool sheep were bred in the alpine pastoral area of Qilian Mountains in the early 1980s, meeting the excellent characteristics of good adaptability to the alpine pastoral area, the wool quality meeting the requirements of the wool spinning industry, and stable genetic performance [1,2]. As a new breed of fine wool meat sheep bred in the 21st century, which are suitable for grazing, semi-grazing, semi-house feeding, and house feeding in alpine pastoral areas, Tianhua mutton sheep inherit not only the wool production characteristics of Gansu alpine fine wool sheep, but also the meat performance of South Africa mutton Merino sheep [3].

Rumen histomorphology is generally described by the height of the rumen epithelial papillae, cuticle thickness, muscle layer thickness, and other related characteristics [4]. The height and width of the rumen epithelial papillae are important indicators to evaluate rumen health and functional efficiency. Rumen epithelial papillae are structures on the rumen mucosa, and their presence helps to improve the surface area of the rumen, thereby enhancing its digestive capacity and absorption efficiency. A variety of microorganisms, collectively known as the epimural microbiota, inhabit the rumen epithelium. These microbes are believed to strictly interact with the epithelial cells and influence the efficiency of the ruminal digestion [5]. The rumen epithelium of ruminants not only prevents external substances from penetrating into tissues, but also performs functions of energy metabolism, nutrient absorption and transport, and immune barrier [6,7]. The mucus layer contained in the cuticle of the rumen epithelium is capable of interacting with keratinized cells to form a molecular or ionic barrier, which plays a large role in the regulation of the active transport and trans-epithelial diffusion of ions in the gastric mucosa [8]. In addition to the rumen tissue morphology, volatile fatty acids and rumen microflora also play an important barrier role in the ruminant rumen. In the process of feeding and rumination, ruminants can secrete a large amount of weakly alkaline saliva, which flows into the rumen as a good buffer to neutralize carbohydrate fermentation and produce a large amount of volatile fatty acids. VFAs have important roles in ruminants, such as providing energy and maintaining a normal rumen environment [9], and the rumen can maintain normal function and stabilize the internal environment when the concentration of rumen VFAs is appropriate and the composition is reasonable [10,11]. Shen et al. [12] demonstrated that the diet–SCFA axis maintains host–microbe homeostasis by promoting the diversification of the extra-mural microbiota and maintaining the integrity of the rumen epithelium in healthy animals, as well as by enhancing the immune barrier activity in animals with a low rumen pH. Similarly, studies have shown that rumen short-chain fatty acids (SCFAs) are important signaling molecules for the rumen microbiota and regulate a variety of physiological functions in the rumen [13].

The unique growth environment in the rumen of ruminants creates suitable conditions for the growth and reproduction of microorganisms, and the rumen, as a unique digestive organ of ruminants, is inhabited by a complex and large microbial community, which plays a key role in the health, growth performance, and immune performance of ruminants [14]. The rumen epithelial barrier and microbial interactions were found to regulate host rumen growth and development, immune function, and metabolic levels [15]. Microorganisms in the rumen include four major groups of microorganisms: protozoa, fungi, archaea, and bacteria. Bacteria are the dominant group of microorganisms, with the highest abundance of Firmicutes and Bacteroidetes, most of which are related to carbohydrate metabolism, while Bacteroidetes is mainly related to starch digestion and metabolism. In addition, there are Fibrobacterota, Gracilibacteria, Spirochaetota, Euryarchaeota, and other bacteria, which jointly maintain and restrict the stability and balance of the rumen environment [16]. The structure and function of rumen flora were found to be influenced by diet, environmental factors such as seasonal temperature and humidity variations, breed, sex and age, and the physiological condition of the host, with the diet composition, nutrient levels, and feed intake considered to be the key factors influencing rumen microbial communities [17]. Guo et al. [18] found that Tibetan goat males and females have different fermentation and metabolic abilities when adapting to the plateau environment, with great differences in rumen microbiota composition and structure. Lin et al. [19] showed that small-tailed Han sheep were more enriched in Proteobacteria, γ-proteobacteria, Aeromonadales, and Succinivibrionaceae genera compared to Boer goats. Similarly, King et al. [20] found that by comparing the rumen flora structure of Jersey cattle and Holstein cows, the content of individual bacteria in the rumen of Jersey cattle was significantly higher than that of Holstein cows. Huang et al. [21] found that the rumen flora composition of Tibetan sheep was better than that of Gansu alpine fine wool sheep. Chang et al. [22] found that the rumen flora richness and diversity of small-tailed Han sheep and Dorper sheep were significantly lower than those of Tibetan sheep. The above studies indicated that the relative abundance of rumen communities in different breeds was different due to the genetic differences in animal breeds. This study analyzed the differences in rumen tissue morphology, rumen VFA content, and rumen microflora between Tianhua mutton sheep and Gansu alpine fine wool sheep under the same grazing environment. We aimed to find out the internal reasons why Tianhua mutton sheep are more suitable for breeding in the high cold grazing area than Gansu alpine fine wool sheep from the perspective of rumen development, and to find a more theoretical basis for the development of the mutton sheep industry in the high cold area and the popularization and application of Tianhua mutton sheep.

## 2. Materials and Methods

### 2.1. Sample Collection

The sample collection site was located in Duolong Village (Wuwei, China), at an altitude of 2800 m. The winter grasslands in this region are home to hardy plants, such as herbs (*Silphium perfoliatum* L., winter pasture 70 rye) and shrubs. The experimental group was raised and managed under the traditional natural grazing mode, and no supplementary feeding was carried out during grazing, and the sheep were free to eat and drink. In this study, 12 sheep (6 Tianhua mutton sheep and 6 Gansu Alpine fine wool sheep) aged 6 months with good health status were selected to be grazed and raised together. After the sheep were slaughtered in December 2023, rumen contents and tissue samples were collected and loaded into 5 mL freezing tubes, which were placed in a liquid nitrogen tank and brought back to the laboratory. A small piece of tissue was cut from the middle part of the rumen ventral sac and placed in a 15 mL centrifuge tube filled with 4% paraformaldehyde. After all the samples were brought back to the laboratory, H&E staining, VFA determination, and 16S rRNA full-length sequencing were performed. The rumen contents of Tianhua mutton sheep (Trumen-group) were numbered as Trumen-Trumen6, and the rumen contents of Gansu alpine fine wool sheep (Grumen-group) were numbered as Grumen1–Grumen6.

### 2.2. H&E Stain

H&E staining was performed by referring to the methods of Wang et al. [23]. The rumen samples that had been fixed by 4% paraformaldehyde were paraffin embedded and made into paraffin sections with a thickness of 5 µm, which were stained by HE staining, and the morphology and structure of each section were observed under a 4× light microscope. The rumen papilla height, papilla width, muscular layer thickness, and cuticle thickness in the field of view were measured with Image-ProPlus 6.0 software, and images were captured.

### 2.3. Determination of Rumen VFAs

VFAs were determined by gas chromatography (HP6890N, Agilent Technologies, Wilmington, DE, USA) with 2-ethylbutyric acid (2EB) as the internal standard. The rumen content samples were treated with reference to the method proposed by Liu et al. [8]. After the samples were centrifuged at 5000 rpm for 10 min, 1 mL of supernatant was added into a 1.5 mL centrifuge tube, and then 0.2 mL of 25% metaphosphate deproteinization solution (containing 2 g/L of internal standard 2EB) was added, mixed well, and centrifuged at 8000 rpm for 10 min. The supernatant was sucked up with a disposable syringe, and then filtered into a 2 mL brown sample bottle using an organic phase filter nozzle with a 0.22 μm tip, and stored at −20 °C. The supernatant was sucked up with a disposable syringe, put on a 0.22 μm organic phase filter tip, filtered into a 2 mL brown sample bottle, and stored at −20 °C for determination. Column: HP19091N-213 capillary column (Agilent, USA). Chromatographic conditions: the detector temperature of the gas chromatograph was 280 °C, the temperature of the injection port was 250 °C, and the temperature of the column was programmed to increase as follows: the temperature was kept at 60 °C for 2 min, then increased to 140 °C at 10 °C/min without retention, and then increased to 170 °C at 3 °C/min.

### 2.4. DNA Extraction from Gastrointestinal Contents

Microbial DNA was extracted from rumen chyme using TGuide S96 Magnetic Bead Method Soil/Fecal Genomic DNA Extraction Kit (Tiangen Biotechnology (Beijing, China) Co., Ltd.) according to the instructions, total RNA was eluted with AE lysis solution, concentration and purity were tested by NanoDrop, and amplified bands were detected by labchip and agarose electrophoresis. The total amount and quality of the samples met the standards for the next step of amplification and sequencing.

### 2.5. 16S rRNA Full-Length Sequencing

After DNA quality control, specific primers with Barcode were synthesized according to the full-length primer sequences (primer sequences: 27F:AGRGTTTGATYNTGGCTCAG; 1492R:TASGGHTACCTTGTTASGACTT), and the products were purified, quantified, and homogenized to form a sequencing library (SMRT Bell). The constructed libraries were first subjected to quality control, and the qualified libraries were sequenced by PacBio Sequel II. The sequencing work was performed by Biomarker Biotechnology Co., Ltd. (Beijing, China).

### 2.6. Sequencing Data Processing

The downstream data of PacBio Sequel II were in bam format, and the CCS files were exported through smrtlink analysis software (version 8.0). The data of different samples were identified and converted into fastq format according to Barcode sequence, and length filtering was performed to remove chimerism and obtain Effective CCS. The Effective CCS sequence was classified into OTUs (later uniformly called Feature) by clustering/de-noising, and its species classification was obtained according to the sequence composition of Feature. Alpha diversity analysis was used to analyze the species diversity within each sample [24], and the Ace, Chao1, Shannon, and Simpson indices of each sample were counted to draw the dilution curves and rank abundance curves of all samples [25]. Beta diversity analysis was used to compare the differences in community composition and structure of all samples of the two groups of sheep species, and NMDS analysis, PCA analysis, PCoA analysis, and PLS-DA analysis of the two groups of sheep were drawn according to the binary_jaccard distance algorithm [26,27]. The matplotlib-v1.5.1 database of the python2 software was used to analyze the species distribution of the two groups’ phyla and the genus level, and the ete3, v3.0.0b35 database of the python2 software was used to plot the degree of species composition difference in all samples at the phylum and genus level. We also used the lefse database of python2 software to find the biomarkers with statistical differences among different groups through the significance analysis of differences between groups [28]. Finally, based on the 16S rRNA sequencing results, we used picrust2 (2.3.0) software to conduct inter-group functional prediction analysis through KEGG and COG databases [29].

### 2.7. Statistical Analysis

Rumen volatile fatty acid content, tissue morphological indexes, and rumen microbial α diversity index were preliminarily sorted by Excel 2019. SPSS 26.0 statistical software was used to conduct one-way ANOVA, and the results were expressed as mean ± standard deviation. Spearman’s correlation test was used to analyze the correlation between rumen microbes and tissue morphology and VFA content. The larger the correlation coefficient, the stronger the correlation between the variables. A Spearman’s analysis greater than 0.5 indicates a strong positive correlation between the two variables and a Spearman’s analysis less than −0.5 indicates a strong negative correlation between the two variables.

## 3. Results

### 3.1. Differences in Volatile Fatty Acids in Rumen Contents of Tianhua Mutton Sheep and Gansu Alpine Fine Wool Sheep

The contents of acetic acid, propionic acid, and butyric acid in the rumen were measured to compare the differences in the volatile fatty acids in the rumen of two sheep. As shown in Table 1, there were no significant differences in propionic acid, butyric acid, and A:P between the rumen contents of the Tianhua mutton sheep and Gansu alpine fine wool sheep (*p* > 0.05). The acetic acid and total acid contents of the rumen contents were significantly different between the Tianhua mutton sheep and Gansu alpine fine wool sheep (*p* < 0.05).

### 3.2. Differences in Rumen Histomorphology between Tianhua Mutton Sheep and Gansu Alpine Fine Wool Sheep

As shown in Table 2, the rumen epithelial development of the two types of fine wool sheep was somewhat different, as shown by the fact that the rumen papilla height of the Tianhua mutton sheep was significantly higher than that of the Gansu alpine fine wool sheep (*p* < 0.05), and the rumen papilla width, stratum corneum thickness, and muscular thickness of the Tianhua mutton sheep and Gansu alpine fine wool sheep were not significantly different from each other (*p* > 0.05).

### 3.3. Alpha Diversity Analysis of Rumen Flora in Tianhua Mutton Sheep and Gansu Alpine Fine Wool Sheep

As shown in Figure 1A,B, the number of features of the two sheep’s bacterial flora were obtained by clustering or denoising, and 7326 feature numbers were found in the rumen of the Gansu alpine fine wool sheep and 8557 feature numbers were found in the rumen of the Tianhua mutton sheep, and the number of features common to the rumen of the two types of fine wool sheep were 5441. The dilution curve and the Shannon index curve both flatten out (Figure 1C,D), indicating that the sampling number was reasonable, the amount of sequencing data were large enough, and the species distribution was relatively even. The α-diversity analyses of the two types of fine wool sheep (Table 3) showed that the Ace, Shannon, and Chao1 indices of the Tianhua mutton sheep were significantly higher than those of the Gansu alpine fine wool sheep (*p* < 0.01), and the differences in the Simpson indices of the two types of fine wool sheep were not significant. The above results indicated that the abundance and diversity of the rumen flora of the Tianhua mutton sheep were higher than those of the Gansu alpine fine wool sheep.

### 3.4. Beta Diversity Analysis of Rumen Flora in Tianhua Mutton Sheep and Gansu Alpine Fine Wool Sheep

Beta diversity analysis was carried out by four methods, and the PCoA analysis as a linear model showed that the rumen flora structure of the Tianhua mutton sheep and Gansu alpine fine wool sheep were significantly different, in which R = 0.741, *p*-value = 0.003 (Figure 2A). NMDS analysis as a non-linear model also reflected the above results, where stress = 0.0375. Thus, the results indicated that the samples within the two groups were close together, and the samples between the two groups could be significantly separated, with significant differences between the two fine wool sheep’s rumen flora (Figure 2B). PLS-DA adopted the classical partial least squares regression model as a supervised discriminant analysis method, which also showed that the rumen flora of the Tianhua mutton sheep and Gansu alpine fine wool sheep were significantly different (Figure 2C). PCA analysis showed that the contribution value of the first principal component to sample difference was 26.78%, and the contribution value of the second principal component to sample difference was 14.63% (Figure 2D).

### 3.5. Analysis of the Difference in Species Composition of the Rumen Flora of Tianhua Mutton Sheep and Gansu Alpine Fine Wool Sheep

As shown in Figure 3A,B, for the rumen microorganisms at the phylum level, Firmicutes had the highest abundance, 55.96% and 60.29% for the Tianhua mutton sheep and Gansu alpine fine wool sheep, respectively. Bacteroidota abundance was next at 39.15% and 35.94% in the Tianhua mutton sheep and Gansu alpine fine wool sheep, respectively. Verrucomicrobiota abundance accounted for 1.73% and 1.16% in the Tianhua mutton sheep and Gansu alpine fine wool sheep, respectively. At the genus level (Figure 3C,D), *Christensenellaceae_R_7_group* had the highest abundance, with 15.48% and 16.94% in the Tianhua mutton sheep and Gansu alpine fine wool sheep, respectively. *Rikenellaceae_RC9_gut_group* had the second-highest abundance, with 15.18% and 13.99% in the Tianhua mutton sheep and Gansu alpine fine wool sheep, respectively.

### 3.6. LEfSe Analysis of Rumen Flora in Tianhua Mutton Sheep and Gansu Alpine Fine Wool Sheep

LefSe analysis screened three differential species, including one at the order level, one at the family level, and one at the genus level (Figure 4A). As is shown in Figure 4B, at the genus level, *Prevotella* was the high-abundance differential species in the Tianhua mutton sheep, and at the family level and the order level, lachnospiraccac and Lachnospirales were the high-abundance differential species in the Gansu alpine fine wool sheep (Figure 4C).

### 3.7. Predictive Analysis of Metabolic Pathways and Functions of Rumen Flora in Tianhua Mutton Sheep and Gansu Alpine Fine Wool Sheep

Species composition information obtained from the 16S sequencing data of the two fine wool sheep was compared by Picrust2 software to hypothesize the functional differences between the two fine wool sheep. When using the KEGG database (Figure 5A), the results indicated that the Tianhua mutton sheep were different in translation, ribosome structure and biosynthesis, cell wall/membrane/biosynthesis, replication, reorganization and repair, and energy production and conversion were enriched in the biometabolic pathways higher than those of the Gansu alpine fine wool sheep, and the Gansu alpine fine wool sheep were enriched in biometabolic pathways higher than those of the Tianhua mutton sheep in transcription, carbohydrate transport, and metabolism. When using the COG database (Figure 5B), the results indicated that the Tianhua mutton sheep were enriched in biometabolic pathways higher than those of the Gansu alpine fine wool sheep in energy metabolism, replication and repair, and translation. The enrichment of the membrane transport and signaling pathways in the Gansu alpine fine wool sheep was higher than that in the Tianhua mutton sheep.

### 3.8. Correlation Alysis of Dominant Bacteria with Histological Morphology and VFAs Indexes

The top 10 microorganisms in the rumen phylum and genus level were selected for correlation analysis with the tissue morphology and VFA content (Figure 6). It turned out that the rumen papilla height was positively correlated with the content of *Ruminococcus*, *NK4A214_group*, Patescibacteria, Verrucomicrobiota, Desulfobacterota, Spirochaetota, and other bacteria. Acetic acid, propionic acid, butyric acid, and total acid were positively correlated with Patescibacteria and negatively correlated with *Succiniclasticum*. In addition, acetic acid was positively correlated with Verrucomicrobiota, Cyanobacteria, and *Ruminococcus*, and negatively correlated with *Acetitomaculum* and *Succiniclasticum*. Proteobacteria was inversely proportional to propionic acid and rumen papilla width, and positively proportional to A:P.

## 4. Discussion

### 4.1. Differences in Volatile Fatty Acids in the Rumen Contents of Tianhua Mutton Sheep and Gansu Alpine Fine Wool Sheep

Tianhua mutton sheep and Gansu alpine fine wool sheep, as special ruminants in alpine pastures, ferment natural pasture grasses through the rumen and produce energy for the body. Among them, the fermentation products, volatile fatty acids (VFAs), are the most important energy source for sheep in alpine pastures in winter. In this study, we measured the rumen VFA content of Gansu alpine fine wool sheep and Tianhua mutton sheep to investigate the differences in rumen fermentation function and adaptive capacity of these two species of fine wool sheep to cope with the nutrient-deficient environment of alpine pastures in winter. The results showed that the acetic acid and total acid contents of Tianhua mutton sheep were higher than those of Gansu alpine fine wool sheep. It has been reported that excessive intake of roughage with a high fiber content will increase the acetic acid content in the rumen [30,31]. A large amount of acetic acid produced by rumen fermentation in sheep is oxidized through the tricarboxylic acid cycle for energy supply or used for fatty acid synthesis to maintain energy intake in winter [32,33]. Acetic acid and propionic acid are the main sources of energy for the rumen papillae, and they produce ATP through oxidation for use by the rumen papillae and other cells. Propionic acid is known to be the most effective activator of rumen papillae growth [34,35]. Similarly, the lower the A:P ratio, the more energy-efficient the feed may be, depending on the specific needs of the animal and the other components of the feed [36]. Therefore, we speculated that the rumen development level, digestive ability, and adaptive ability of Tianhua mutton sheep were better than those of Gansu alpine fine wool sheep.

### 4.2. Differences in Rumen Histomorphology between Tianhua Mutton Sheep and Gansu Alpine Fine Wool Sheep

VFAs produced by rumen fermentation are absorbed into the host organism for energy supply through the rumen epithelium [37], and the development of the rumen epithelium also plays an important role in terms of the rumen barrier and nutrient absorption [38]. Rumen muscular layer development is mainly stimulated by the physical stimulation of feed [39], and in this study, there was no significant difference in the rumen muscular layer thickness between the Tianhua mutton sheep and Gansu alpine fine wool sheep, which may be caused by the consistent growth environment and diet. In addition, as a defense barrier in the rumen environment, the stratum corneum is often in direct contact with the rumen contents, and there is no significant difference in the rumen defense barrier ability between the two fine wool sheep. The height and width of the rumen papillae are usually used as a criterion for evaluating the digestive and absorptive capacity of rumen nutrients and an important indicator for evaluating rumen development [40]. In this experiment, we found that the rumen papilla height of Tianhua mutton sheep was significantly higher than that of Gansu alpine fine wool sheep (*p* < 0.05), and there was no significant difference in the rumen papilla width between Tianhua mutton sheep and Gansu alpine fine wool sheep (*p* > 0.05). Therefore, we guessed that the rumen nutrient digestion and absorption capacity of Tianhua mutton sheep was higher than that of Gansu alpine fine wool sheep.

### 4.3. Differences in the Rumen Flora Structure of Tianhua Mutton Sheep and Gansu Alpine Fine Wool Sheep

Rumen microorganisms play an important role in the process of the environmental adaptation and nutrient digestion and absorption of sheep, and the community structure of rumen microorganisms of different breeds is significantly different [40]. Cheng et al. [41] compared the rumen flora of Hu sheep, Tan sheep, and Dorper sheep by principal component analysis, and found that the three sheep breeds could be separated and had a very different community structure. Cholewinska et al. [42] conducted microbiological analysis on three kinds of sheep that were raised in the same environment and fed the same diet, and found that there were significant differences in the level of bacterial population between different breeds. Studies have shown that sheep breeds with better rumen flora richness and diversity have a better adaptability to the environment. In this study, the rumen flora of Tianhua mutton sheep had more characteristic numbers than those of Gansu alpine fine wool sheep, and the richness and diversity of the rumen flora of Tianhua mutton sheep were significantly higher than those of Gansu alpine fine wool sheep, suggesting that Tianhua mutton sheep had a better adaptability than Gansu alpine fine wool sheep. Chang et al. [22] found that the rumen flora richness and diversity of small-tailed Han sheep and Dorper sheep were significantly lower than those of Tibetan sheep. The above studies indicated that the relative abundance of rumen communities in different breeds was different due to the genetic differences in animal breeds. Both PCoA analysis and NMDS indicated significant differences in the rumen flora structure between Tianhua mutton sheep and Gansu alpine fine wool sheep, and the samples within the two fine wool sheep groups were in close proximity. Chang et al. [22], through principal coordinate analysis (PCoA) and non-metric multidimensional scale (NMDS) analysis, showed that the microbiome composition of Tibetan sheep was significantly different from that of the other three sheep breeds (*p* < 0.01). The rumen microbiota of ruminants at the phylum level was dominated by Firmicutes and Bacteroidota. The sum of Firmicutes and Bacteroidota was more than 95.11% in both fine wool sheep in this experiment. Firmicutes contain many fiber-decomposing bacteria, which are related to fiber degradation and fat deposition in the rumen [42,43]. Bacteroidota has the ability to degrade plant polysaccharides and is mainly involved in the degradation process of starch [44,45], and they help digest and absorb carbohydrates. The dominant genera in Gansu alpine fine wool sheep and Tianhua mutton sheep in this experiment were *Christensenellaceae_R_7_group* and *Rikenellaceae_RC9_gut_group*. *Christensenellaceae_R_7_group* is associated with host health and is involved in proteolytic metabolic processes in feed [46].

LEfSe analysis enables comparisons between two or more subgroups, looking for biomarkers that are statistically different between subgroups. A total of three differential species were screened in this study, one at the order level, one at the family level, and one at the genus level. At the genus level, *Prevotella* is a highly abundant and differentiated species of Tianhua mutton sheep, and at the family level and order level, lachnospiraccac and Lachnospirales are highly abundant and differentiated species of Gansu alpine fine wool sheep. *Prevotella* contributes to the breakdown of protein and carbohydrate foods, while Lachnospiraccac and Ruminococcaceae have been strongly associated with recurrent disease, adverse reactions to anti-tumor necrosis factor therapy, and relapses after surgical intervention in patients with Crohn’s disease [47,48]. The KEGG and COG databases are commonly used to study the extent of alterations in metabolic function that occur in sample microbial communities in response to environmental changes. Both the KEGG and COG in this experiment showed that Tianhua mutton sheep were enriched in biometabolic pathways such as replication and repair and translation than Gansu alpine fine wool sheep, in which the microbial replication and repair metabolic pathways were closely related to bacterial growth [19,49]. The diversity and abundance of the rumen flora of the Tianhua mutton sheep in this study were higher than those of the Gansu alpine fine wool sheep, suggesting that the good adaptation of Tianhua mutton sheep to alpine pastures may be related to the replication and repair of rumen microorganisms, and the expression of translational metabolic pathways.

Rumen microorganisms act together with VFAs and tissue morphology to influence rumen development. *Ruminococcus* is primarily responsible for the breakdown of resistant starch and plays an important role in rumen fiber catabolism and has a significant impact on other microbial populations in the rumen. *NK4A214_group* affects the stability of the rumen environment and the efficiency of nutrient conversion by participating in the structure of the microbial community and metabolic activities in the rumen, which in turn has an impact on the development of the rumen. Verrucomicrobiota members are able to break down cellulose in the rumen, releasing small molecular compounds that can be utilized by other microorganisms and the host organism. Spirochaetota is able to participate in the breakdown of complex polysaccharides and proteins and produce B-complex vitamins in the rumen. Therefore, we speculate that *Ruminococcus*, *NK4A214_group*, Patescibacteria, Verrucomicrobiota, Desulfobacterota, Spirochaetota, and Cyanobacteria are beneficial to rumen development.

Of course, the above results indicate that Tianhua mutton sheep have certain advantages in rumen development under the same grazing environment, which may be an important factor for Tianhua mutton sheep to adapt to the growth of alpine grazing areas and improve mutton production, but the specific mechanism still needs to be further studied.

## 5. Conclusions

In conclusion, the contents of acetic acid, total acid, and papilla height of rumen metabolites of the growing ewes of Tianhua mutton sheep were significantly higher than those of Gansu alpine fine wool sheep. In addition, the content of beneficial bacteria such as Verrucomicrobia, Bacteroidota, *Rikenellaceae_RC9_gut_group*, and *Prevotella* of Tianhua mutton sheep was higher than that of Gansu alpine fine wool sheep. From the perspective of rumen development, these results explain the possible reasons why Tianhua mutton sheep are more suitable for living in alpine pastoral areas than Gansu alpine fine wool sheep, and provide a certain theoretical basis for the promotion and application of Tianhua mutton sheep and healthy breeding.

## Figures and Tables

**Figure 1 animals-14-01259-f001:**
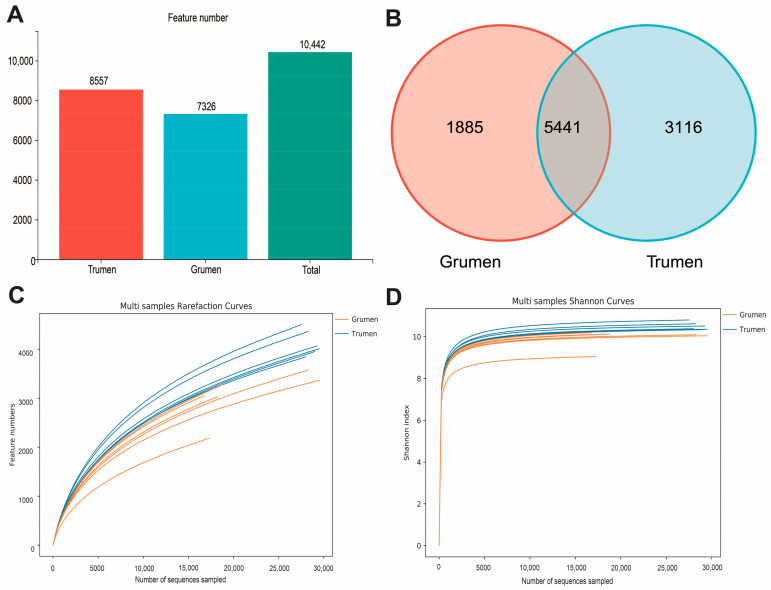
Alpha diversity analysis. (**A**) Feature number bar chart; (**B**) Venn diagram; (**C**) dilution curves; (**D**) Shannon exponential curve. Grumen: rumen bacterial flora of the Gansu alpine fine wool sheep; Trumen: rumen bacterial flora of the Tianhua mutton sheep.

**Figure 2 animals-14-01259-f002:**
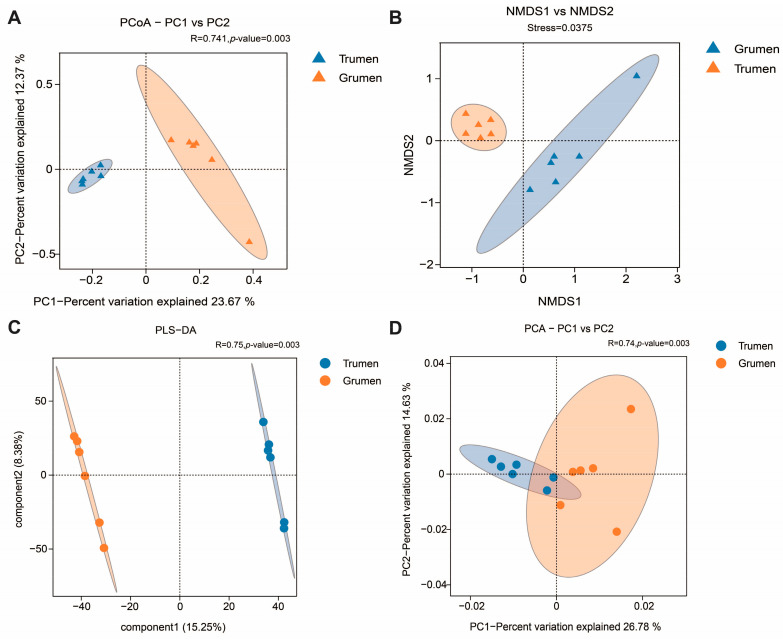
Beta diversity analysis. (**A**) PCoA analysis; (**B**) NMDS analysis; (**C**) PLS-DA analysis; (**D**) PCA analysis. Grumen: rumen bacterial flora of the Gansu alpine fine wool sheep; Trumen: rumen bacterial flora of the Tianhua mutton sheep.

**Figure 3 animals-14-01259-f003:**
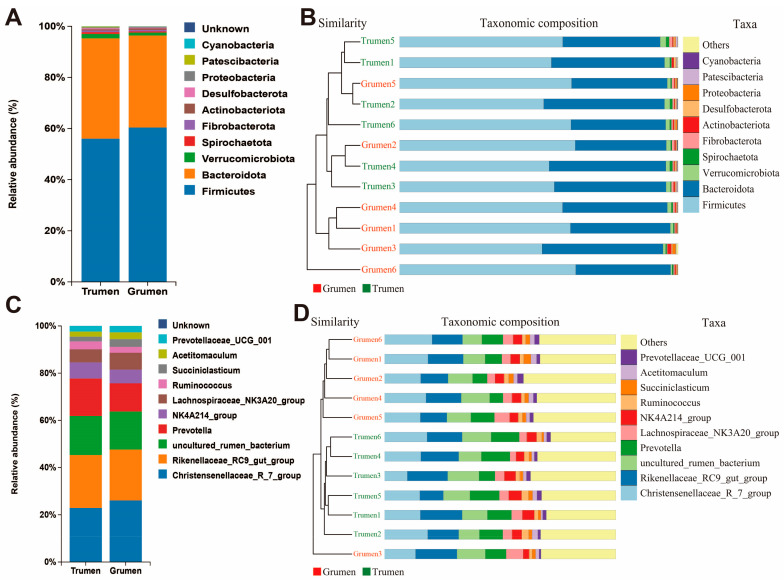
Analysis of differences in species composition of rumen flora. (**A**) Composition of two sheep species at the phylum level; (**B**) UPGMA tree diagrams of all samples at the phylum level; (**C**) composition of two sheep species at the genus level; (**D**) UPGMA tree of all samples at the genus level. Grumen: rumen bacterial flora of the Gansu alpine fine wool sheep; Trumen: rumen bacterial flora of the Tianhua mutton sheep.

**Figure 4 animals-14-01259-f004:**
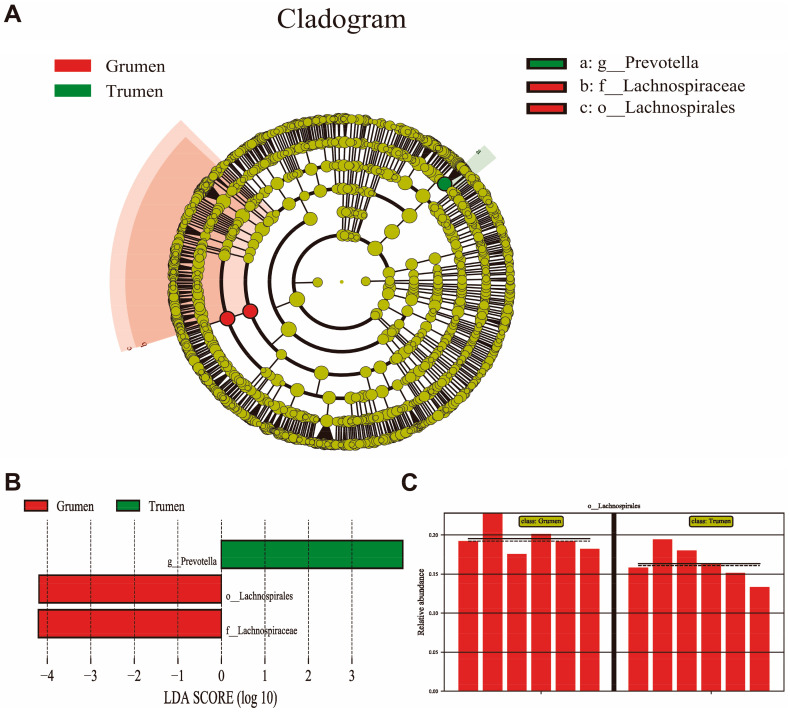
Significance analysis of differences between groups. (**A**) LEfSe analysis; (**B**) LDA score chart; (**C**) Lachnospirales bacteria difference map; The solid lines in each bar chart represent the mean value of the samples in that group, and the dashed lines represent the median value of the relative abundance of the samples in that group. Grumen: rumen bacterial flora of the Gansu alpine fine wool sheep; Trumen: rumen bacterial flora of the Tianhua mutton sheep.

**Figure 5 animals-14-01259-f005:**
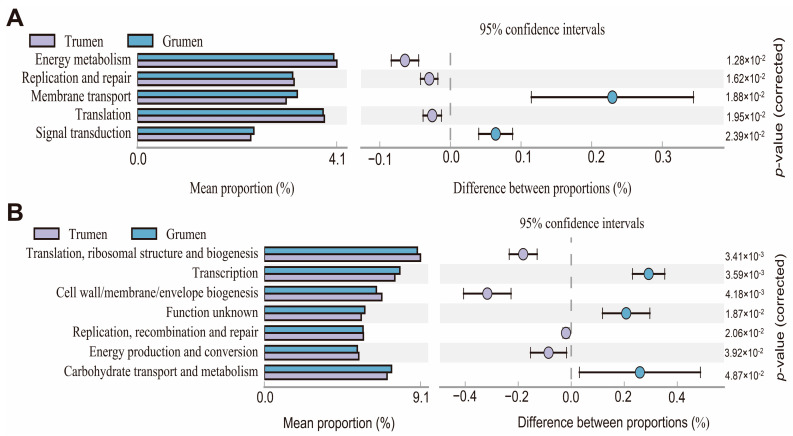
Prediction of metabolic pathway and function of rumen flora. (**A**) Analysis of KEGG metabolic pathway differences. (**B**) Analysis of KEGG metabolic pathway differences. Grumen: rumen bacterial flora of the Gansu alpine fine wool sheep; Trumen: rumen bacterial flora of the Tianhua mutton sheep.

**Figure 6 animals-14-01259-f006:**
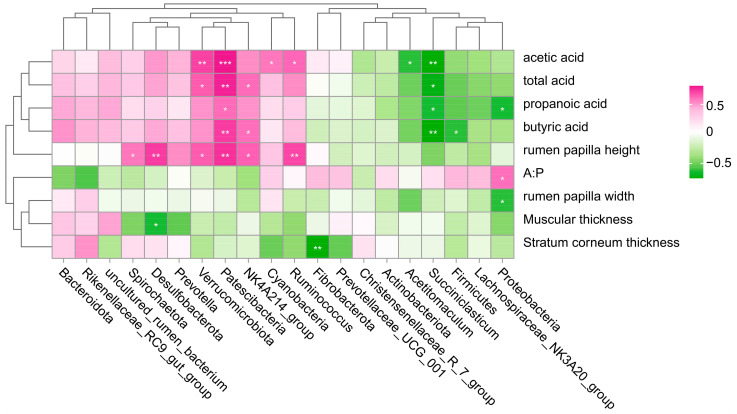
Correlation analysis of dominant bacteria with histological morphology and VFA indexes. * *p* < 0.05; ** *p* < 0.01; *** *p* < 0.001.

**Table 1 animals-14-01259-t001:** Differences in volatile fatty acids in rumen contents of Tianhua mutton sheep and Gansu alpine fine wool sheep.

Items	Tianhua Mutton Sheep	Gansu Alpine Fine Wool Sheep	*p*-Value
acetic acid, mmol/L	43.47 ± 1.51 ^a^	36.38 ± 1.55 ^b^	0.010
Propanoic acid, mmol/L	15.74 ± 1.55	11.85 ± 2.35	0.123
Butyric acid, mmol/L	14.88 ± 0.35	12.07 ± 2.22	0.151
Total acid ^1^, mmol/L	84.53 ± 3.82 ^a^	71.56 ± 5.03 ^b^	0.044
A:P ^2^	2.74 ± 0.20	3.17 ± 0.51	0.329

Means in the same row with different superscripts differ (*p* < 0.05). ^1^ Total acids include acetic, propionic, butyric, valeric, isobutyric, and isovaleric acids. ^2^ A:P, acetic acid/propanoic acid.

**Table 2 animals-14-01259-t002:** Differences in rumen epithelial development in Tianhua mutton sheep and Gansu alpine fine wool sheep.

Items	Tianhua Mutton Sheep	Gansu Alpine Fine Wool Sheep	*p*-Value
Rumen papilla height, μm	1715.17 ± 43.28 ^a^	1279.80 ± 144.79 ^b^	0.015
Rumen papilla width, μm	522.97 ± 145.39	462.03 ± 39.98	0.598
Stratum corneum thickness, μm	91.53 ± 15.80	84.17 ± 5.25	0.565
Muscular thickness, μm	1246.00 ± 180.92	1240.23 ± 11.95	0.966

Means in the same row with different superscripts differ (*p* < 0.05).

**Table 3 animals-14-01259-t003:** A diversity analysis of Tianhua mutton sheep and Gansu alpine fine wool sheep.

Items	Tianhua Mutton Sheep	Gansu Alpine Fine Wool Sheep	*p*-Value
Feature	4134.00 ± 234.11 ^a^	3094.17 ± 443.57 ^b^	0.001
ACE	5754.15 ± 286.24 ^a^	4788.52 ± 431.60 ^b^	0.002
Chao1	5858.27 ± 318.77 ^a^	4578.07 ± 583.13 ^b^	0.002
Simpson	1.00 ± 0.01	1.00 ± 0.01	0.141
Shannon	10.50 ± 0.17 ^a^	9.94 ± 0.40 ^b^	0.016
PD_whole_tree	230.18 ± 8.12 ^a^	185.54 ± 14.68 ^b^	0.001
Coverage	0.95 ± 0.01	0.94 ± 0.01	0.279

Means in the same row with different superscripts differ (*p* < 0.05).

## Data Availability

The datasets presented in this study can be found in online repositories. The names of the repository/repositories and accession number(s) can be found below: https://www.ncbi.nlm.nih.gov/bioproject/PRJNA1085975 (accessed on 9 March 2024).

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
