# Peer review of "Rumen Development of Tianhua Mutton Sheep Was Better than That of Gansu Alpine Fine Wool Sheep under Grazing Conditions"

_animals, 2024, doi:10.3390/ani14091259_

Round 1

Reviewer 1 Report

Comments and Suggestions for Authors

I have specific comments and suggestions as follows.

L20 HE, VFA spell out when mention in the first time.

L32,34,35 suggest deleting 1)2)3).

L90 change to "pH".

L121-126 move to discussion section.

L135 describe details in animal breed, sex, age, 

weight, feeding and feeding management. 

L140 "HE" "VFA" spell out when appear first time.

L144-194 add the related methods references.

Table 1 total acid1 => delete "1"

L 214 delete row 214.

L228 delete "Bul".

L336-351 add discussion on the roles

 of C2 & C3 on rumen muscular papilla 

development with related references.

L370 check reference citation.

Author Response

Dear Reviewer

       Thank you very much for your constructive comments on our article. Based on your suggestions, we have made extensive revisions to the previous manuscript. We uploaded the revised version, and our changes are as follows:

  1. L20 HE, VFA spell out when mention in the first time.

We sincerely thank you for your careful reading, and we have added the full names of HE and VFA in L20.   

Hematoxylin Eosin (HE) staining, Volatile Fatty Acids (VFA)

  1. L32,34,35 suggest deleting 1)2)3).

Thanks to your suggestion, we have removed these three subheadings. See L32, 34,35 for details.

  1. L90 change to "pH".

We are very sorry for our careless mistake. The "PH" of the full text has been changed to "pH".

  1. L121-126 move to discussion section.

Thank you for your suggestion, but considering that there is also relevant language in the discussion section, we have revised this sentence. Let's change it to "This study analyzed the differences of rumen tissue morphology, rumen VFAs content and rumen flora structure between Tianhua mutton sheep and Gansu alpine fine wool sheep under the same grazing environment, aiming to find out the internal reasons why Tianhua mutton sheep is more suitable for breeding in the high cold grazing area than Gansu alpine fine wool sheep, and to find more theoretical basis for the development of meat sheep industry in the high cold area and the promotion and application of Tianhua mutton sheep.".

  1. L135 describe details in animal breed, sex, age, weight, feeding and feeding management.

Thank you for your careful inspection. We apologize for our careless behavior. We added key factors such as animal breed, sex and age in the sample collection section.

The sample collection site was located in Duolong Village (China, Wuwei) , at an altitude of 2800m. The winter grasslands in this region are home to hardy plants, such as herbs (Silphium perfoliatum L, winter pasture 70 rye) and shrubs. The experimental group was raised and managed under the traditional natural grazing mode, and no supplementary feeding was carried out during grazing, and the sheep were free to eat and drink. In this study, 12 sheep (6 Tianhua mutton sheep and 6 Gansu Alpine fine wool sheep) aged 6 months with good health status were selected in December 2023.

  1. L140 "HE" "VFA" spell out when appear first time.

Thank you for your suggestion, but we have already added the full names of HE and VFA in line 20, so we do not think it is necessary to add them here.

  1. L144-194 add the related methods references.Table 1 total acid1 => delete "1"

Thanks for your suggestion, we have supplemented the relevant references in the material method; The label in Table 1 is to explain the composition of total acid, but because we made a mistake in the label in the table, it caused confusion in reading. We fixed the error and added A note about A:P.

  1. L 214 delete row 214.

       Thanks for your careful check, we have deleted the empty line.

  1. L228 delete "Bul".

       Thanks for your careful reading, we have deleted "Bul".

  1. L336-351 add discussion on the roles of C2 & C3 on rumen muscular papilla development with related references.

         Thank you very much for your constructive comments, we supplemented the role of C2 and C3 in rumen muscular papilla development, and supplemented the relevant references.

        Acetic acid and propionic acid are the main sources of energy for the rumen papillae, and they produce ATP through oxidation for use by the rumen papillae and other cells. Propionic acid is known to be the most effective activator of rumen papillae growth [34,35];

  1. L370 check reference citation.

   Thank you for your reminding, but we are very sorry that we did not find the error. At the same time, we have checked the format of all the references in the full text and modified them to the correct format.

Reviewer 2 Report

Comments and Suggestions for Authors

This paper discusses the rumen characteristics between the Tianhua mutton sheep and the Gansu alpine fine wool sheep to better elucidate how the Tianhua sheep are more adaptable to the high mountain plains of the Qilian Mountains. Generally, this paper is prepared well and seems to fit a knowledge gap related to the utilization of these breeds within the stated region. I commend the authors for the breadth of the study and all of the data collection, analysis, and manuscript preparation. There are, however, a few questions and corrections to the current manuscript.

Broad questions / comments:

- Double check that reference numbers are appropriately boxed [##], as there are a few spots where this is not done (Line 116 & 370).

- Also, there is some inconsistencies on capitalization and spacing of "P values" which needs to be corrected.

- How old were the sheep at harvest? What kind of diet were they on prior to harvest?

- There are a number of references in the introduction & discussion about adaptability to grazing these mountain plains, however no information is given about the sheep other than they are harvested for sample collection. Especially for VFA & Microbiology work in ruminants, the diet is a key piece of information to put the expectations into perspective, so this is absolutely critical to the paper!

- Generally, be mindful of defining abbreviations prior to their first use and starting sentences with abbreviations should be avoided if at all possible.

- The tables need legends so that they can be more easily read as stand-alone items.

Specific items:

- Line 60 - 63, this sentence does not make sense as to what you are trying to tell your reader.

- Line 90, "PH" should be "pH".

- Line 139, convert the ";" to a "."

- Line 145, remove "fixed" from between rumen and samples

- For the HE staining (Line 144 - 150) & VFA determination (Line 151 - 166), please use a reference for the procedure, either manufacturer specifications or previous research.

- Line 169, add the manufacturer of the Extraction kit

- Table 1: Add legend for superscript 1 description; Also add superscript 2 for A:P to define this parameter

- Lines 215  - 222 & Table 2: Is "nipple" referring to "papilla"? If so, why was nipple used as it seems like an unnecessarily confusing term.

- Table 2, add units to each item on the table & legend for "a,b" superscript.

- Line 251, the phrase "sheep can't get together" in this sentence does not make sense

- Table 3, Gansu "hair" sheep, should be "wool" sheep. Add legend with superscript description

- Line 325 & 326, is there a reference for this statement about energy efficiency? Wouldn't this be largely impacted by the diet of the animal, which is not described in the Materials & Methods

- Line 336 - 337, since the growth environment & diet are not described in this paper, I am not sure that this description or conclusion is supported.

- Line 337 - 341, is a repeat of lines 334 - 337.

- In References, # 22 & # 33 are the repetitive.

Comments on the Quality of English Language

Generally the English is good. Few minor issues are present.

Author Response

Dear Reviewer

Thank you very much for your recognition of our articles and your constructive comments. Based on your suggestions, we have made extensive revisions to the previous manuscript. We uploaded the revised version, and our changes are as follows:

Broad questions / comments:

1.Double check that reference numbers are appropriately boxed [##], as there are a few spots where this is not done (Line 116 & 370).

We sincerely thank you for your careful reading, the references are all framed

2.Also, there is some inconsistencies on capitalization and spacing of "P values" which needs to be corrected.

Thank you for your careful check, the size and spacing of P-values have been uniformly modified.

3.How old were the sheep at harvest? What kind of diet were they on prior to harvest?

Thank you very much for your constructive comments and we have added relevant information. The Tianhua mutton sheep and Gansu alpine fine wool sheep selected for this experiment were all 6-month-old grazing sheep in the same area. We restricted feeding (stopping feeding for 12 hours) the day before collection.

4.There are a number of references in the introduction & discussion about adaptability to grazing these mountain plains, however no information is given about the sheep other than they are harvested for sample collection. Especially for VFA & Microbiology work in ruminants, the diet is a key piece of information to put the expectations into perspective, so this is absolutely critical to the paper!

Thank you for your reminder that we have added key factors about animal species, gender, age, diet and so on in the material method.

The sample collection site was located in Duolong Village (China, Wuwei) , at an altitude of 2800m. The winter grasslands in this region are home to hardy plants, such as herbs (Silphium perfoliatum L, winter pasture 70 rye) and shrubs. The experimental group was raised and managed under the traditional natural grazing mode, and no supplementary feeding was carried out during grazing, and the sheep were free to eat and drink. In this study, 12 sheep (6 Tianhua mutton sheep and 6 Gansu Alpine fine wool sheep) aged 6 months with good health status were selected in December 2023.

5.Generally, be mindful of defining abbreviations prior to their first use and starting sentences with abbreviations should be avoided if at all possible.

Thanks for your comments, we have added the full name before using the abbreviation for the first time.

Hematoxylin Eosin (HE) staining, Volatile Fatty Acids (VFA) assay

6.The tables need legends so that they can be more easily read as stand-alone items.

Thanks for your careful review, we have added the legend below the table.

Specific items:

  1. Line 60 - 63, this sentence does not make sense as to what you are trying to tell your reader.

Thank you for your reminder, we originally wanted to indicate the relevant background of the breeding breed of Tianhua mutton sheep, but it does not seem to have much significance, so we deleted this sentence.

  1. Line 90, "PH" should be "pH".

      Thank you for your careful inspection. We are very sorry for our careless mistake. We've changed the PH to pH.

3.Line 139, convert the ";" to a "."

Thank you for your careful inspection, we have ";" Change to "." .

4.Line 145, remove "fixed" from between rumen and samples

Thank you for your careful inspection, and we are very sorry for our careless mistake. We deleted the word.

5.For the HE staining (Line 144 - 150) & VFA determination (Line 151 - 166), please use a reference for the procedure, either manufacturer specifications or previous research.

Thanks for your suggestion, we have added relevant references on HE staining and VFA determination.

6.Line 169, add the manufacturer of the Extraction kit

Thanks for your suggestion, we added the manufacturer specification.

Tiangen Biotechnology (Beijing, China) Co., Ltd.

7.Table 1: Add legend for superscript 1 description; Also add superscript 2 for A:P to define this parameter

We sincerely thank you for your careful reading, we superscript 1, and add the definition for A: P.

8.Lines 215-222 & Table 2: Is "nipple" referring to "papilla"? If so, why was nipple used as it seems like an unnecessarily confusing term.

Thank you for your careful inspection, and we are sorry for our careless error. The location should be "papilla", and we have replaced all "Nipple" in the text.

9.Table 2, add units to each item on the table & legend for "a,b" superscript.

Thanks for your careful review, we have added the units of all indicators in Table 2.

10.Line 251, the phrase "sheep can't get together" in this sentence does not make sense

Thanks to your suggestion, we have deleted the sentence "sheep can't get together".

11.Table 3, Gansu "hair" sheep, should be "wool" sheep. Add legend with superscript description

Thank you for your careful check, and we are very sorry for our carelessness. The "hair" in the article should all be "wool". In addition, we have added the explanation of a and b below the table.

12.Line 325 & 326, is there a reference for this statement about energy efficiency? Wouldn't this be largely impacted by the diet of the animal, which is not described in the Materials & Methods

We sincerely thank you for your careful reading, we have revised this sentence, and added the relevant references.

Similarly, the lower the A:P ratio, the more energy efficient the feed may be, depending on the specific needs of the animal and the other components of the feed [36].

13.Line 336 - 337, since the growth environment & diet are not described in this paper, I am not sure that this description or conclusion is supported.

Thank you for your constructive comments. We are very sorry for not explaining clearly the breeding environment and other background. Both types of sheep are raised on the Duolong Village ranch, where they have been raised together since birth. We have added two sheep-related backgrounds to the material methods section.

14.Line 337 - 341, is a repeat of lines 334 - 337.

Thank you for your careful check, we are very sorry for our carelessness, we have deleted the repeated sentence.

15.In References, # 22 & # 33 are the repetitive.

Thanks for your careful review, we have checked all references and removed duplicates.

Reviewer 3 Report

Comments and Suggestions for Authors

Title

Comparison of rumen healthy development between Tianhua mutton sheep and Gansu alpine fine wool sheep

The title could be changed to one that points more precisely the objective of the study.

The work was focused on investigating the differences of rumen tissue morphology, volatile fatty acid content, and rumen microflora between Tianhua mutton sheep and Gansu alpine fine wool sheep under the same grazing conditions. 

Even when this is an interesting work, the writing requires major corrections, including the discussion and conclusions, as well as to mention the limitations and recommendations for future works on the subject.

Comments:

Simple summary: No comments

Abstract: Reconsider the conclusions.

Keywords: No comments

Introduction:

Line 116 and 370: Check the spaces and the citations Huang et al. 21 and Chang et al. 33

Lines 121-127: This text seems the conclusion of the paper.

In this study, the differences of rumen tissue morphology, rumen contents VFAs and rumen flora structure between Tianhua mutton sheep and Gansu alpine fine wool sheep were analyzed under the same grazing environment, which revealed that Tianhua mutton sheep was more suitable for local production and rumen microbial barrier mechanism suitable for house feeding than Gansu alpine fine wool sheep. It provides more theoretical basis for the development of sheep industry and the popularization and application of Tianhua mutton sheep in the alpine region.

Please reconsider the text and replace with the hypothesis.

Materials and methods

Lines 141-143: The description in these lines could be omitted, since it doesn’t seem to be relevant.

The rumen contents of Tianhua mutton sheep (T-group) were numbered as Trumen1-Trumen6, and the rumen contents of Gansu alpine fine wool sheep (G-group) were numbered as Grumen1-Grumen6.

Line 174: Separate 2.6 from 16 in 2.616.

Lines 197-200: Specify which variables were statistically analyzed.

Sections 2.5, 2.6, and 2.7 describe in a very summarized manner the process. It would be desirable that these sections explain the methods in more detail.

Results

Line 214: What does the section marked as “3.1.1. Subs 1.Total acids include acetic, propionic, butyric, valeric, isobutyric and isovaleric acids” refer to?

Table 2: Change the word “nipple” to “rumen papilla”.

Line 228: What does BUL stand for?

Line 236: In the sentence “indicating that the sampling number was reasonable”, please explain what do you mean with “reasonable”.

Avoid duplicating the information described in the results, figures and tables.

Figures 1-5: Add to the figure description the meanings of Grumen and Trumen.

Figure 3: Check the whole figure “Analysis of differences in species composition of rumen flora. (a) phylum level; (b) genus level”. It has four parts (a-d) but in the description only two appear (a and b). Besides, the description for b is incorrect.

Discussion:

Lines 327-328: The statement “Therefore, we concluded that Tianhua mutton sheep have good intake, high feed utilization and good adaptability to alpine pastures” has not enough arguments to be sustained.

Lines 335-340: The result on muscular layer thickness are duplicated.

Line 349: The P-value showed is (P<0.05) but it should be (P>0.05) because it is not significant.

Lines 349-351: About the statement “we concluded that the rumen nutrient digestion and absorption capacity of Tianhua mutton sheep was higher than that of Gansu alpine fine wool sheep”, the absorption capacity could be due to the height of the rumen papilla; however, this is relative since there is no digestibility variable to validate this conclusion.

Even when the differences were obtained to the phylum, genus, and species levels, it must be taken into account that the microbial ecosystem of the rumen works jointly and every bacterial type has multiple associations with other microorganisms, resulting in the production indicators for the VFAs, which are the main source of energy available for the ruminant.

The discussion needs to be refocused to the results obtained and the limitations of this study should be mentioned because no production or growth variables are shown that could support that the adaptation of the sheep to alpine pasture rearing environments is related to an improved growth or meat/wool production.

Conclusion:

The conclusion needs to be reformulated as many of the information is repeated and there is no detail as to what was the hypothesis. Another option is to take into account other variables to have validated results for use in future research. In any case, it should be specified how this information could boost the wool production.

The previous publication by the authors could be used to add arguments to the present work, given that this one focuses on the rumen and the previous one focused on the small intestine. Those results are necessary and can be used to make a more complete evaluation.

Li, D., Liu, Z., Duan, X., Wang, C., Li, Q., & Ma, Y. (2024). Differences in intestinal barrier function between Tianhua mutton sheep and Gansu alpine fine wool sheep (Ovis aries).

References:

Please homogenize the format being used for all the references.

Author Response

Dear Reviewer

Thank you very much for your constructive comments on our article. Based on your suggestions, we have made extensive revisions to the previous manuscript. We uploaded the revised version, and our changes are as follows:

Title

Thanks for your important suggestion, we have revised the title of the paper to "Tianhua mutton sheep had better rumen development than Gansu alpine fine wool sheep".

Abstract

Thanks for your constructive comments, we have made some amendments to the conclusion of the abstract.

 In general, there were some similarities between Tianhua mutton sheep and Gansu Alpine fine wool sheep in rumen tissue morphology, rumen fermentation ability and rumen flora structure. However, Tianhua mutton sheep had better performance in these aspects, especially in rumen acetic acid content, rumen papillae height and beneficial bacteria content. These differences may be one of the reasons why Tianhua mutton sheep is more suitable for growing in alpine pastoral areas than Gansu alpine fine wool sheep.

Introduction:

1.Line 116 and 370: Check the spaces and the citations Huang et al. 21 and Chang et al. 33

Thank you for your careful check. We are sorry for our low-level errors. We have supplemented the quotation format of these two references and revised all the references.

2.Lines 121-127: This text seems the conclusion of the paper.

Thanks for your constructive comments, we have revised this sentence.

This study analyzed the differences of rumen tissue morphology, rumen VFAs content and rumen flora structure between Tianhua mutton sheep and Gansu alpine fine wool sheep under the same grazing environment, aiming to find out the internal reasons why Tianhua mutton sheep is more suitable for breeding in the high cold grazing area than Gansu alpine fine wool sheep, and to find more theoretical basis for the development of mutton sheep industry in the high cold area and the popularization and application of Tianhua mutton sheep.

Materials and methods

1.Lines 141-143: The description in these lines could be omitted, since it doesn’t seem to be relevant.

Thank you for your valuable advice, this sentence is our mistake, the two sheep groups should be Trumen and Grumen, the sample names should be Tru1-Tru6,Gru1-Gru6 respectively. In addition, we are afraid that readers will be confused if the groups or samples in the pictures are not described, so we think this sentence is necessary.

2.Line 174: Separate 2.6 from 16 in 2.616.

Thanks for your careful check, we have corrected the error.

2.6. 16S rRNA full-length sequencing

3.Lines 197-200: Specify which variables were statistically analyzed.

Thanks for your constructive comments, we have revised the relevant statistical method in Section 2.8.

Rumen volatile fatty acid content, tissue morphological indexes and rumen microbial α diversity index were preliminarily sorted by Excel. SPSS 26.0 statistical software was used to conduct one-way ANOVA, and the results were expressed as mean ± standard deviation. P < 0.05 indicated that the difference was significant and had statistical significance.

4.Sections 2.5, 2.6, and 2.7 describe in a very summarized manner the process. It would be desirable that these sections explain the methods in more detail.

We sincerely thank you for your careful reading of the 16S rDNA full-length sequencing we describe in detail in parts 2.5-2.7.

Results

1.Line 214: What does the section marked as “3.1.1. Subs 1.Total acids include acetic, propionic, butyric, valeric, isobutyric and isovaleric acids” refer to?

Thank you for your careful reading. It is our wrong labeling that makes you feel confused. We have labeled 1 in table 1. Considering that total acid might need to be described, we have added a table note below Table 1 to better describe total acid.

2.Table 2: Change the word “nipple” to “rumen papilla”.

Thank you for your careful check, we have misdescribed this noun. We have revised all the "nipple" in the article to "rumen papilla".

3.Line 228: What does BUL stand for?

Thanks for your careful reading, we have removed the BUL error.

4.Line 236: In the sentence “indicating that the sampling number was reasonable”, please explain what do you mean with “reasonable”.

Thanks for your careful reading, we describe this view because in the dilution curve, when the curve tends to be flat, it means that the number of samples taken is reasonable, and more samples will only produce a few new species. So we write here "indicates a reasonable sample size", and here it is reasonable to say that a set of 6 samples is sufficient for later analysis.

  1. Avoid duplicating the information described in the results, figures and tables.

Thanks for your constructive comments, we have revised the description of sample name and group name in Materials and Methods.

Figure 3. Analysis of differences in species composition of rumen flora. (A) Composition of two sheep species at the phylum level; (B) UPGMA tree diagrams of all samples at phylum level; (C) Composition of two sheep species at the genus level; (D) UPGMA tree of all samples at the genus level.

Discussion:

1.Lines 327-328: The statement “Therefore, we concluded that Tianhua mutton sheep have good intake, high feed utilization and good adaptability to alpine pastures” has not enough arguments to be sustained.

Thanks to your constructive comments, we have revised this description.

Therefore, we speculated that the rumen development level, digestive ability and adaptive ability of Tianhua mutton were better than that of Gansu alpine fine wool sheep.

2.Lines 335-340: The result on muscular layer thickness are duplicated.

Thanks for your careful reading, we have deleted the duplicate parts.

3.Line 349: The P-value showed is (P<0.05) but it should be (P>0.05) because it is not significant.

Thank you for your careful reading, we are sorry for our low-level error, we have corrected this error.

4.Lines 349-351: About the statement “we concluded that the rumen nutrient digestion and absorption capacity of Tianhua mutton sheep was higher than that of Gansu alpine fine wool sheep”, the absorption capacity could be due to the height of the rumen papilla; however, this is relative since there is no digestibility variable to validate this conclusion.

Thanks for your constructive comments, this statement is not accurate enough, we will revise this sentence to "we guessed that the rumen nutrient digestion and absorption capacity of Tianhua mutton sheep was higher than that of Gansu alpine fine wool sheep.".

The discussion needs to be refocused to the results obtained and the limitations of this study should be mentioned because no production or growth variables are shown that could support that the adaptation of the sheep to alpine pasture rearing environments is related to an improved growth or meat/wool production.

Thanks for your guidance, we have added the limitations and suggestions of this experiment in the last part of the discussion.

Of course, the above results indicate that Tianhua mutton has certain advantages in rumen development under the same grazing environment, which may be an important factor for Tianhua mutton to adapt to the growth of alpine grazing areas and improve mutton production, but the specific mechanism still needs to be further studied.

Conclusion:

  1. The conclusion needs to be reformulated as many of the information is repeated and there is no detail as to what was the hypothesis. Another option is to take into account other variables to have validated results for use in future research. In any case, it should be specified how this information could boost the wool production.

Thanks for your constructive comments, we have redescribed the conclusion. In addition, Tianhua mutton sheep is a new breed bred to improve local mutton production. Compared with Gansu alpine fine wool sheep in actual production, Tianhua mutton sheep has a high adaptability to alpine cold areas. The purpose of this article is to find out the internal reasons why Tianhua mutton sheep is more suitable for breeding in alpine pastoral areas from the perspective of sheep rumen.

In conclusion, the contents of acetic acid, total acid and papilla height of rumen metabolites of the growing ewes of Tianhua mutton sheep were significantly higher than those of Gansu alpine fine wool sheep. In addition, the content of beneficial bacteria such as Verrucomicrobia, Bacteroidota, Rikenellaceae_RC9_gut_group and Prevotella of Tianhua mutton sheep was higher than that of Gansu alpine fine wool sheep. These results explored the excellent characteristics of the rumen of Tianhua mutton sheep to adapt to the life in the alpine pastoral area, and provide a certain theoretical basis for the popularization and application of Tianhua mutton sheep and healthy breeding.

References:

1.Please homogenize the format being used for all the references.

Thanks to your valuable suggestions, we have revised all the reference formats so that they are unified.

Round 2

Reviewer 2 Report

Comments and Suggestions for Authors

I would like to commend the authors for their hard work in conducting the project, as well as preparing and revising this manuscript. Following the revisions, I see no issues barring publication.

Comments on the Quality of English Language

No major issues.

Author Response

Dear Reviewer

Thank you very much for your significant contributions to our articles, and thank you for your high recognition of our work. We asked other teachers to help us revise the language again.

I wish you a happy life and success in your work.

Best regards,

Youji Ma

Reviewer 3 Report

Comments and Suggestions for Authors

The work was focused on investigating the differences of rumen tissue morphology, volatile fatty acid content, and rumen microflora between Tianhua mutton sheep and Gansu alpine fine wool sheep under the same grazing conditions. 

The document was improved according to the previous revision, so only some minor comments remain. Requires improve conclusions according to the objective of the study

Author Response

Dear Reviewer

Thank you very much for your important contribution to our article, we have revised the introduction and conclusion. We added descriptions of the height and width of the rumen papillae, and deleted descriptions of the basal layer.

L69-L87  

Rumen histomorphology is generally described by the height of rumen epithelial papillae, cuticle thickness, muscle layer thickness ,and other related characteristics [4]. The height and width of rumen epithelial papillae are important indicators to evaluate rumen health and functional efficiency. Rumen epithelial papillae are structures on the rumen mucosa, and their presence helps to improve the surface area of the rumen, thereby enhancing its digestive capacity and absorption efficiency; Rumen epithelial papillae are primarily made up of epithelial cells that secrete a variety of digestive enzymes that help break down cellulose and other nutrients in food [5]. The rumen epithelial of ruminants, as a physical barrier of the contents of the lumen, can resist the intrusion of various substances into the tissue, and has functions such as energy metabolism, absorption and transport, and immune barrier [6,7]. The mucus layer contained in the cuticle of the rumen epithelium is capable of interacting with keratinized cells to form a molecular or ionic barrier, which plays a large role in the regulation of active transport and trans-epithelial diffusion of ions in the gastric mucosa [8]. In addition to the rumen tissue morphology, volatile fatty acids ,and rumen microflora also play an important barrier role in the ruminant rumen. In the process of feeding and rumination, ruminants can secrete a large amount of weakly alkaline saliva, which flows into the rumen as a good buffer to neutralize carbohydrate fermentation and produce a large amount of volatile fatty acids.

L127-L134   

This study analyzed the differences of rumen tissue morphology, rumen VFAs content and rumen microflora between Tianhua mutton sheep and Gansu alpine fine wool sheep under the same grazing environment, aiming to find out the internal reasons why Tianhua mutton sheep is more suitable for breeding in the high cold grazing area than Gansu alpine fine wool sheep from the perspective of rumen development, and to find more theoretical basis for the development of mutton sheep industry in the high cold area and the popularization and application of Tianhua mutton sheep.

L448-L457

From the perspective of rumen development, these results explain the possible reasons why Tianhua mutton sheep is more suitable for living in alpine pastoral areas than Gansu alpine fine wool sheep, and provide a certain theoretical basis for the promotion and application of Tianhua mutton sheep and healthy breeding.

I wish you a happy life and success in your work.

Best regards,

Youji Ma
